# Three-Dimensional Numerical Field Analysis in Transformers to Identify Losses in Tape Wound Cores

**DOI:** 10.3390/s24103228

**Published:** 2024-05-19

**Authors:** Dariusz Koteras, Bronislaw Tomczuk

**Affiliations:** Department of Electrical Engineering and Mechatronics, Opole University of Technology, PL-45758 Opole, Poland; b.tomczuk@po.edu.pl

**Keywords:** core losses, medium-frequency transformer, 3D magnetic field analysis, iterative homogenization method (IHM), measurement verification

## Abstract

To find the total core losses in 1-phase medium-frequency transformers, a 3D numerical field analysis was carried out. The proposed numerical modeling was based on the extended iterative homogenization method (IHM) developed by the authors. The achieved calculation results were validated by the corresponding values obtained experimentally, and a reasonably close agreement was obtained.

## 1. Introduction

Many electrical devices were built using cores made of soft ferromagnetic materials. These cores were mainly manufactured either as laminated or as magnetically soft composites [1]. The soft magnetic materials are characterized by main parameters such as magnetic flux density *B*, magnetic field intensity *H*, and core losses *P*, as well as the magnetic permeability. Due to their application in many branches of industry, their properties can be significantly different [2,3,4]. In 2019, oriented steel was mainly used (at 60% share) in the global soft magnetic materials market; Figure 1 [5]. Lately, however, the sheets have been gradually replaced by amorphous ribbons, whereas ferrites, with a 13% share of the market, have taken second place.

The core losses inside magnetic materials are caused by two physical phenomena—eddy currents and hysteresis [6]. Inside the transformer’s laminated core, the magnetic flux density vector is perpendicular to the smallest cross-section of the single sheet in the stack; Figure 2. Under these operating conditions, the eddy current component of the losses can be found using the equation below:(1)$$P_{ec}=\int \frac{E^{2}}{R}dV$$
where *Ε* is the rms value of the electromotive force, *R* is the resistance of the single sheet, and *V* is the volume of the sheet.

Calculation of the formula above leads to a well-known analytical expression for eddy current loss determination [7].
(2)$$P_{ec}=\frac{1}{6}\sigma {\left(\pi fB_{mav}d\right)}^{2}V$$
where σ is the electrical conductivity of the sheet, *f* is the frequency of the magnetic flux density, *B_mav_* is the maximum value of the flux density in the cross-section of a sheet, and *d* is the thickness of the sheet.

Equation (2) concerns the linear model, which has been derived from the saturation-wave concept (SWM) by Wolman and Kaden [8]. In the literature [9,10], it has been proved that it is valid in the case where *B_p_* = *B_m_*. Thus, we have used it for the linear part of the magnetization curve.

The second phenomenon, i.e., hysteresis, has been investigated for many years. It can be described by two models: the Preisach and the Jiles–Atherton. The first model is described using an infinite set of simplest hysteresis operators *ŷ_αβ_*, which represent the hysteresis nonlinearities. Each of these operators can be represented by the rectangular loop shown in Figure 3, [11]. Along with the set of operators *ŷ_αβ_*, we use an arbitrary weight function μ (*α*, *β*).

Using the presented above quantities, the so-called Preisach hysteresis operator $\hat{\Gamma }$ is defined as
(3)$$\hat{\Gamma }u\left(t\right)=\int_{\alpha \geq \beta }^{\;}\mu \left(\alpha ,\beta \right){\hat{\mu }}_{\alpha \beta }u\left(t\right)d\alpha d\beta$$

Applying this model in the numerical analysis of magnetic fields in ferromagnetic materials is widely described in the literature [12,13].

The second model of hysteresis described in the literature is the Jiles–Atherton model. It is based on the total differential susceptibility given in the below equation [14]:(4)$$\frac{dM}{dH}=1-c\frac{M_{an}-M_{irr}}{k\delta -\alpha \left(M_{an}-M_{irr}\right)}+c\frac{dM_{an}}{dH}$$
where *M_an_*—anhysteretic magnetization; *M_irr_*—irreversible magnetization; *δ*—a directional parameter (+1, –1); *k*—coercivity; *H_c_*—parameter; and *α*—coefficient of interdomain coupling in the magnetic material.

Some parameters in this model, like α, *k*, and *c*, are obtained from the experimental data expressed through the form of a hysteresis loop [15]. This model is still used to calculate hysteresis in soft ferromagnetic materials [16]. Below are shown quite simple analytical formulas based on available measurement coefficients, i.e., Richter’s and Steinmetz’s approaches for hysteresis loss calculation [17]:(5)$$P_{hS}=\eta fB_{mav}^{1.6}m$$
(6)$$P_{hR}=\varepsilon \frac{f}{100}B_{mav}^{2}m$$
where *η*, *ε*—empirical coefficients; *f*—frequency of the flux change; B*_mav_*—maximal average value of the flux density in the magnetic core; and *m*—the mass of the core.

There is a significant difference between the core loss values measured and those calculated with numerical field analysis and the analytical expressions. That difference is called the excess (auxiliary) losses component, *P_ex_*. Thus, the total losses can be assumed to consist of three components: eddy current losses (*P_ec_*,), hysteresis ones (*P_h_*), and those *P_ex_*. The first interpretation of the excess losses was given by Pry and Bean who have taken into account the ratio 2 *L*/*d* of the domain size (*L*) and the lamination thickness (*d*) [18]. However, their relationship includes too many simplifications and does not give reasonable accuracy. Below a more precise analytical expression is given in which the coefficients obtained from measurements can be applied. It is related to so-called active magnetic objects (MOs) of ferromagnetic materials [18,19].
(7)$$P_{ax}=\frac{8\cdot B_{mav}\cdot f\cdot m}{\rho }\left(\sqrt{\sigma \cdot G\cdot S\cdot B_{mav}\cdot f\cdot V_{0}}-\frac{{\tilde{n}}_{0}\cdot V_{0}}{4}\right)$$
where *B_mav_*—the maximal average value of the flux density in the core; *S*—the cross-sectional surface perpendicular to the flux direction in the core; *G*—the dimensionless coefficient (usually equal to 0.1356); *ñ*_0_—the limiting number that characterizes the magnetic objects which are statistically independent when *f*→0; and *V*_0_ is dependent on the (*H_ax_*, *ñ*) function when *f* → 0.

Nowadays, many electrical devices operate under medium frequencies, i.e., in the range of tens of kilohertz, which is much higher than the technical frequencies of 50 or 60 Hz. This results in smaller dimensions of the magnetic core geometry. However, for a medium-frequency core, the losses significantly increase. Thus, the proper values of appropriate loss must be predicted for these operating conditions when designing medium-frequency transformers.

For medium-frequency transformers, the analytical expressions concerning the core losses either do not exist or are given for selected frequency values only [2,3]. They are also inconvenient for direct use in numerical analysis, especially in 3D modeling, where instead of core lamination we should consider a solid structure of the magnetic circuit. Thus, in this paper we describe a 3D numerical analysis, based on our IHM [20,21], to calculate the total core losses inside the laminated core of the 1-phase transformer (Figure 4). In this method, the laminated core has been modeled as a solid structure with equivalent parameters such as conductivity *σ_eq_* and relative magnetic permeability *μ_req_*. In this method, the core is treated as a solid geometry with equivalent parameters (relative magnetic permeability and electrical conductivity). In the literature so far, the homogenization method mainly concerns simply laminated packages of grain-oriented silicon steels without measurement verification [22,23]. A significant part of this research concerns the numerical analysis of losses that occur in the transformer windings. For example, the paper in [24] numerically investigates the copper loss of a three-phase transformer dry-type 300 kVA under various geometry designs.

## 2. Numerical Models

### 2.1. Analyzed Objects

This paper presents the results of our research concerning 1-phase transformers with a laminated “*C*”-type core made of two soft ferromagnetic materials; Figure 4.

The core of transformer T1 is made of the grain-oriented silicon steel OS-110 Cut “A” (GOSS), and the core of transformer T2 is made from amorphous ribbons of iron-based METGLAS^®^ 2605SA1 [25,26]. For the 3D field analysis, a Cartesian coordinate system was assumed, as presented in Figure 4. According to it, the main dimensions of both transformers are given in Table 1.

To facilitate the calculations, instead of the layered core, we considered its modular structure, in the “*C*” letter form, of two solid parts. Additionally, to avoid losses in the transformer windings (coils), we used Litz wire. Both windings, with the same number of turns (*N* = 21), were connected in series.

### 2.2. Numerical Model

Our numerical analysis used the Maxwell 3D module from the commercial Ansys simulation package. In this package, the Finite Element Method is implemented using automatic mesh generators. Therefore, interference in the density of the discretization mesh is limited. However, within the Maxwell 3D package, it is possible to refine the mesh on the boundaries of the object and subareas by specifying the length of the elements. In Figure 5a,b, the finite element grids generated for the two transformers are shown. The total number of elements in the grid is equal to 39,158 for the transformer *T*_1_ (with GOSS) and 16,497 for the *T*_2_ amorphous one. The correctness of the simulations is achieved after obtaining the previously assumed energy error. They were executed in four passes for the first transformer and three passes for the second one.

For the 3D calculation, we investigated the eddy current models with the frequency domain. The algorithm of the solver is based on the ***T*–**Ω formula, where ***T*** is an electric vector potential and Ω is a magnetic scalar potential [27,28]. Obviously, the non-conducting regions hold the below equation:(8)$$div\left(grad\Omega \right)=0$$

In the regions with non-zero conductivity, where eddy currents flow, the below equation must be solved:(9)$$\Delta T=j\omega \mu \sigma_{eq}\left(H_{S}+T-grad\Omega \right)$$
where *σ_eq_* is the equivalent conductivity of the solid core and *μ* is the magnetic permeability of it; ***H_S_*** is magnetic field strength excited by the coils.

Below are given, implemented in Ansys software, version R1 from 2020 the relationships for each component of the losses:(10)$$P_{ec}=K_{ec}f^{2}B_{mav}^{2}$$
(11)$$P_{h}=K_{h}fB_{mav}^{2}$$
(12)$$P_{ax}=K_{ax}f^{1.5}B_{mav}^{1.5}$$

To calculate the total core losses based on the proper value of each coefficient, *K_ec_* for eddy currents, *K_h_* for hysteresis and *K_ex_* for excess losses, our modified IHM was applied. A simplified flow chart of that algorithm is given in Figure 6.

Firstly, the *P_TotD_*_, i.e._, total losses of the transformer, and *B_mD_*_, i.e._, the magnetic flux density maximal average value, are assumed. These values, inside the core, are related to the operating conditions of the transformer. At the start of the solver execution, the transformer’s operating conditions and maximal number *n* of iterations are assumed. The algorithm starts with assumed initial values for each coefficient *K_ec_*, *K_h_*, and *K_ax_*. After the magnetic field analysis, we verified the value of magnetic flux density inside the core. When its estimation is correct, i.e., the relative error of the magnetic flux density *ε_Bm_* is lower than the assumed one, the value of the total loss is checked. When this value is appropriate, i.e., the relative error of the total core losses *ε_Bm_* is lower than that assumed, the algorithm stops the iterative process. Finally, when the calculations reach the *n* number of iterations, it will be also stopped to prevent continuous looping.

Our numerical model extends the application of the IHM method described in [16]. In this model, we have assumed the equivalent conductivity of the core (at *σ_eq_* = 700 S/m). This value was valid for the entire core, in which an almost constant value of the magnetic flux density was maintained (Figure 7a,b). It gives a good approximation to the real laminated cores, where each single sheet has a similar *B_mav_* value inside the core. On the other hand, the equivalent relative permeability *μ_req_* was selected in an iterative process to make it possible to obtain the same average value of the magnetic flux density *B_mav_* in the cross-sectional plane halfway up the transformer column.

## 3. Calculation Results and Measured Verification

In our numerical analysis, the investigated transformer was powered by sinusoidal current waves of frequencies ranging from 50 Hz to 400 Hz.

Due to the overheating of the core during measurements at the increased frequency, the tests were carried out for relatively low values of flux density. The presented calculation results of the core losses were given at the value of *B_mav_* = 0.6 T and the excitation current wave frequency of *f* = 400 Hz. The magnetic flux density *B* module is presented in Figure 7a,b by the color bitmaps used for the *B* module distribution inside the core’s *XZ* plane (Figure 4). Note that the maximum values of *B* are in the subregions near the corners of the transformer window. In contrast, the minimum ones are in the outer part of the core corners. In the columns and yokes, the *B* values are almost equal. For the transformer *T*_1_, the *P_ec_* component has been included in the total losses. Thus, in Figure 8, the eddy current *J* distribution is presented inside the *XZ* plane.

Figure 9a,b show components of the total losses for transformer (GOSS) *T*_1_ vs. frequency. In Figure 9a the percentage share for each component of the total losses is presented, whereas Figure 9b gives the values of each component. As can be seen, for the frequency equal to *f* = 50 Hz, the components caused by the eddy currents *dP_ec_* and hysteresis *dP_h_* phenomena have the same percentage share, which equals approximately 30% of the total losses. For the gradually increased value of the frequency, the component *dP_ec_* significantly increases until its percentage share achieves about 80%.

Although the total losses in transformer *T*_2_ are determined correctly, their division for the amorphous package seems to be slightly erroneous. Due to the narrow hysteresis loop of amorphous materials in the considered frequency range, the hysteresis losses in the *T*_2_ transformer are much lower than those in transformer *T*_1_ (from GOSS). Moreover, the losses from eddy currents, calculated by Equation (2), constitute approximately 1% of the total losses [29]. However, the measurement results give higher values. Additionally, the loss components defined by Formulas (10)–(12) also contribute to the power share components imprecisely. These calculations show that auxiliary losses account for almost all of the total ones, which seems slightly strange and will be the subject of our further research.

Figure 10a,b present the measured core losses for the entire scope of our tests. For the frequencies from 50 up to 400 Hz and the magnetic flux density from 0.1 up to 1.2 T, calculations are included for the transformer *T*_1_ (with GOSS core), and the flux density from 0.1 up to 1.0 T for the amorphous object for the transformer *T*_2_ was obtained. It can be noticed that the maximum value of the losses for the amorphous core is about three times lower.

As mentioned above, the total core losses in transformer *T*_1_ (with GOSS) consist of all three components, whereas in amorphous transformer *T*_2_, the *P_ax_* component seems to be a decisive part of the total losses. Thus, Figure 11a,b give the coefficients, calculated with the IHM algorithm, occurring in Equations (10)–(12). However, the measurement results for transformer *T*_2_ give higher values of the hysteresis and eddy current losses than the calculated ones. Moreover, the loss components defined by Formulas (10)–(12) also contribute to the power share components imprecisely. These calculations show that auxiliary losses account for almost all of the total ones, which seems slightly strange and will be the subject of our further research.

The linearity of the *K_ec_* characteristic results from analytical Equation (2) and from that implemented in (10). From the relationships, it is visible that the frequency *f* exponent and the magnetic flux density *B_mav_* are the same value and equal to 2. The decreasing values of the other coefficients result from the fact that the component losses have a lower share in the total losses (Figure 9). Moreover, the values of the frequency f exponent and the flux density *B_mav_* differ in the analytical and the numerically implemented equations. Similarly, in the amorphous transformer (*T*_2_), the graph of the *K_ax_* values results from the fact that the total losses versus the frequency values increase due to the linear approach (Figure 10b), whereas using Equation (12), the losses change non-linearly due to the exponent number, which is equal, i.e., 1.5, vs. the frequency function.

Our numerical analysis was verified by the measurement tests with the 8-bit digital oscilloscope Tektronix MSO 2024B. The simplified diagram of the measurement system is given in Figure 12. Figure 13a,b show the calculated and measured values of the iron losses versus frequency values *f.* Our calculations give the possibility to determine the losses for other frequencies, where the material parameters of solid cores were determined based on the approximation between the nodes of the interactive process, where the total discrepancy between measurements and loss calculations is visible. Figure 13a,b show a comparison of the calculated and measured values of the total losses for both analyzed transformers. The small differences of about 10% between the values given in these figures validate our modeling of the losses in the laminated core.

## 4. Conclusions

In the currently published approaches, the design of electromagnetic devices is supported by 3D numerical field analysis. Proper modeling for each separate sheet, or a strip of it, in the laminated core is almost impossible. Therefore, in this paper, we describe calculation models with solid core geometry. Our iterative numerical approach is based on mathematical modeling with the equivalent parameters, such as the conductivity *σ_eq_* and the relative magnetic permeability *μ_req_*, of the core. In our analysis, described herein, the proper values of the coefficients (*K_ec_*, *K_h_*, and *K_ax_*) for each component of the core losses have been calculated. The correctness of the presented models has been validated by the measurements, and a satisfying convergence of results for both models was reached (Figure 13a,b). Consequently, we recommend the proposed IHM method as applicable in calculations of both GOSS sheets, with their domain structure, and amorphous ribbons.

## Figures and Tables

**Figure 1 sensors-24-03228-f001:**
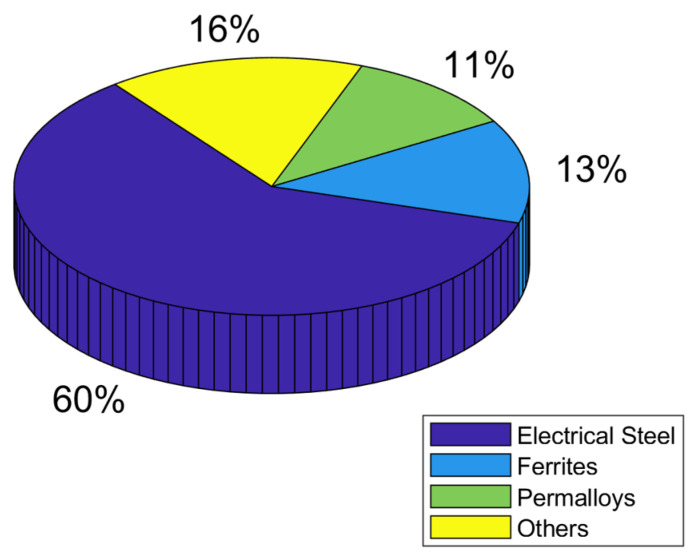
A pie graph for the global soft ferromagnetic materials market share in 2019.

**Figure 2 sensors-24-03228-f002:**
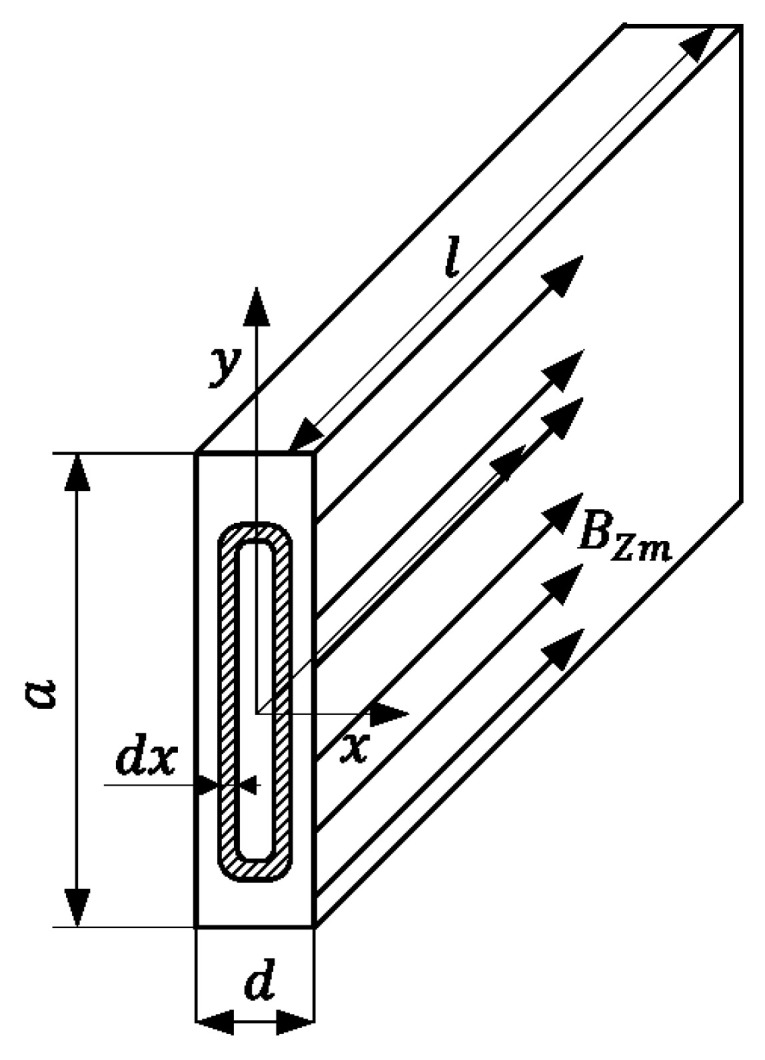
Assumed magnetic field conditions to calculate the eddy current losses component.

**Figure 3 sensors-24-03228-f003:**
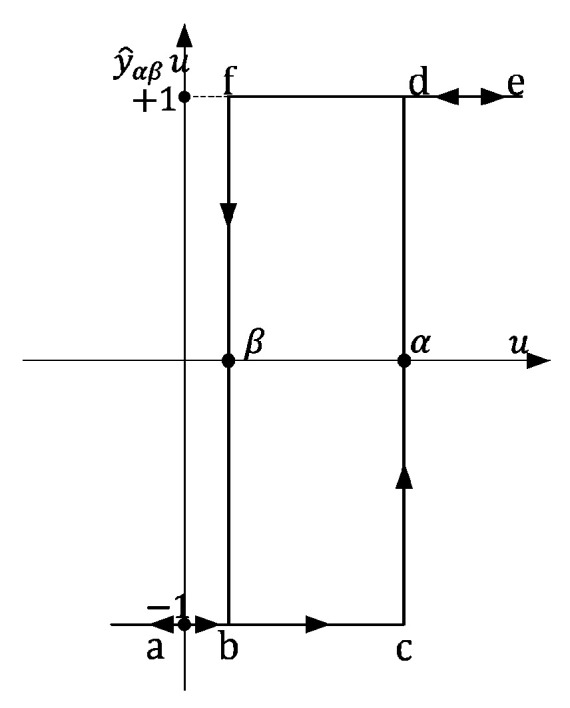
An elementary rectangular loop in the Preisach model.

**Figure 4 sensors-24-03228-f004:**
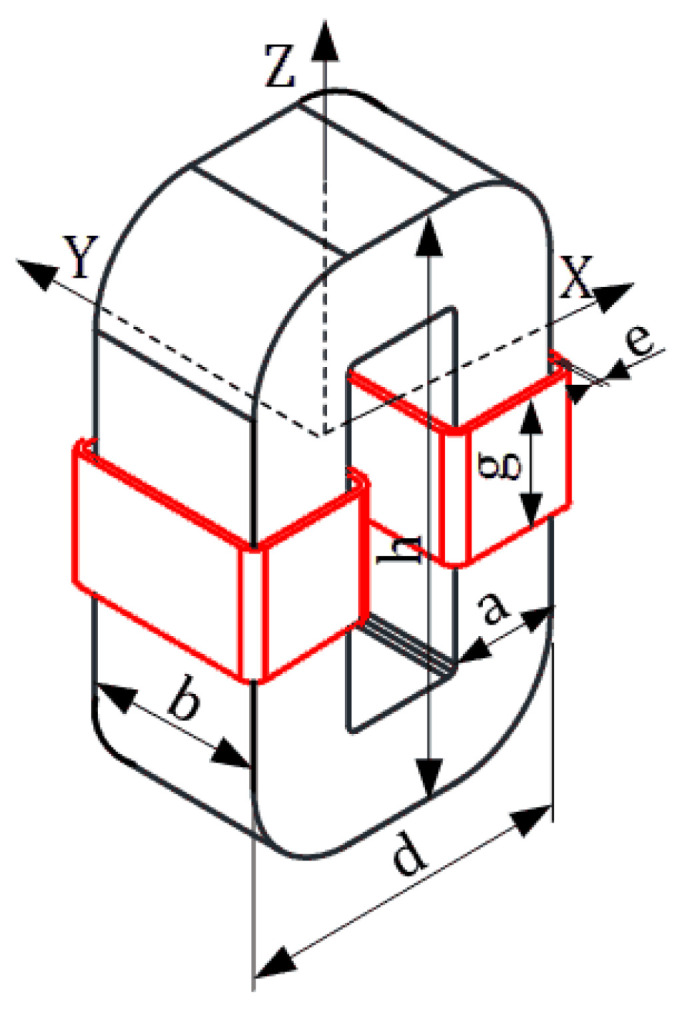
Outline of the analyzed objects.

**Figure 5 sensors-24-03228-f005:**
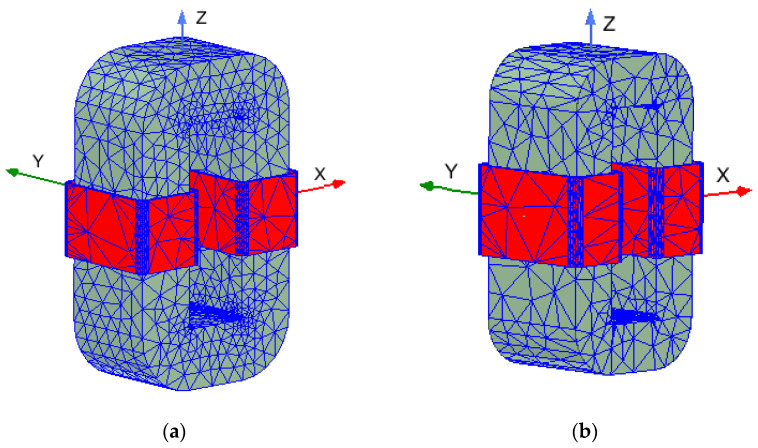
Finite element grids: (**a**) for transformer *T*_1_; (**b**) for transformer *T*_2_.

**Figure 6 sensors-24-03228-f006:**
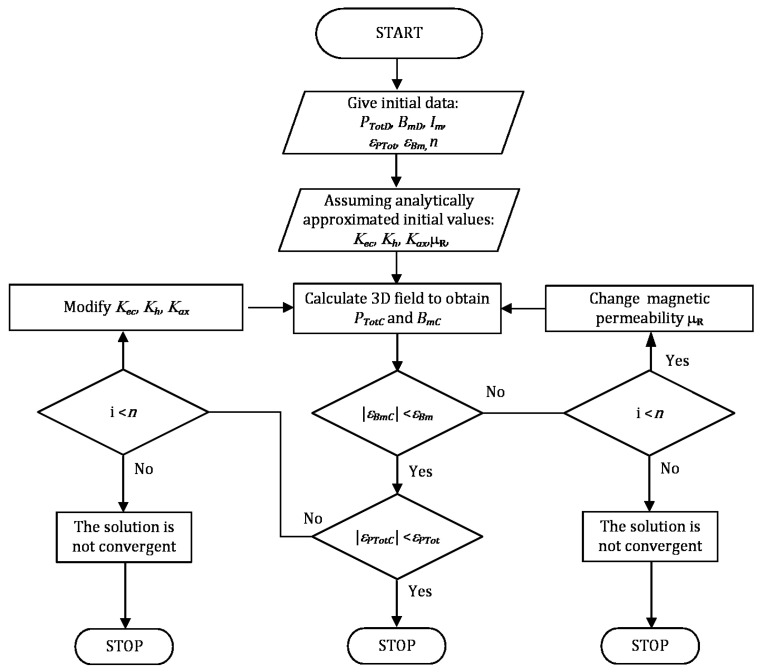
Flow chart of the algorithm for modified IHM.

**Figure 7 sensors-24-03228-f007:**
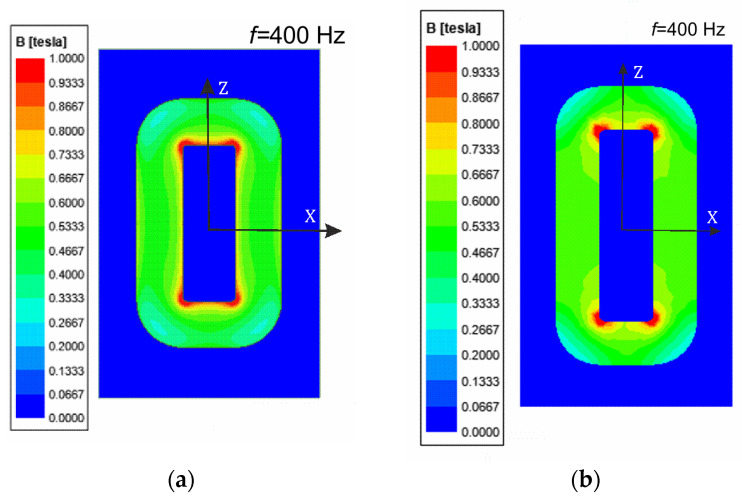
The *B* distribution inside the core on the XZ plane: (**a**) for transformer *T*_1_; (**b**) for transformer *T*_2_.

**Figure 8 sensors-24-03228-f008:**
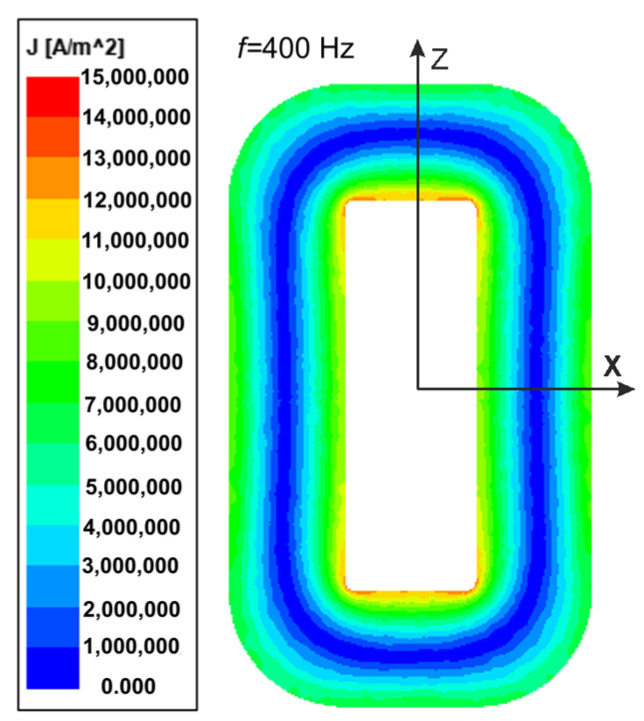
The eddy current density *J* distribution inside the *XZ* plane of the transformer *T*_1_ core.

**Figure 9 sensors-24-03228-f009:**
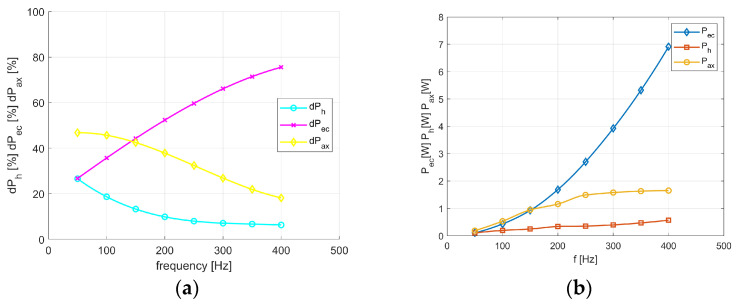
Components of the total losses for transformer *T*_1_ vs. frequency: (**a**) percentage share of the total losses components; (**b**) values of each component.

**Figure 10 sensors-24-03228-f010:**
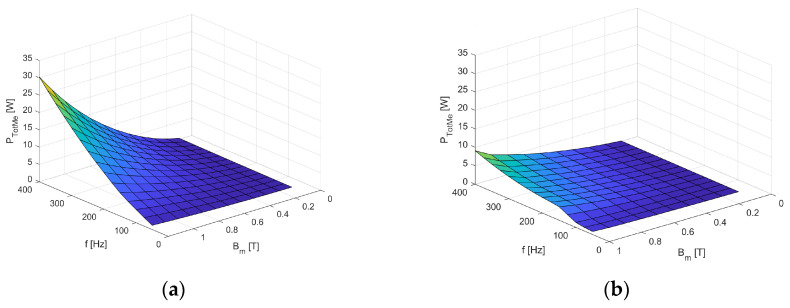
Relationship *P_TotMe_* = *f(f, B_m_)*: (**a**) for transformer *T*_1_; (**b**) for transformer *T*_2_.

**Figure 11 sensors-24-03228-f011:**
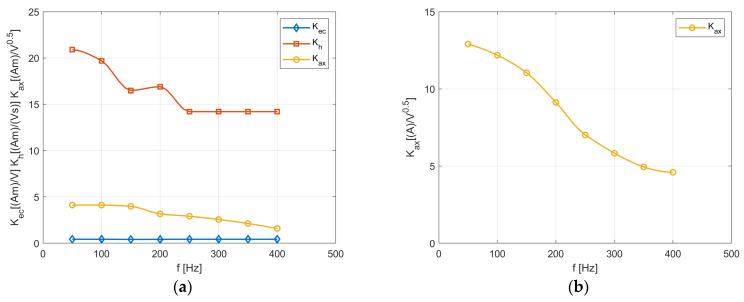
Comparison of the calculated and measured total losses: (**a**) for transformer *T*_1_; (**b**) for amorphous transformer *T*_2_.

**Figure 12 sensors-24-03228-f012:**
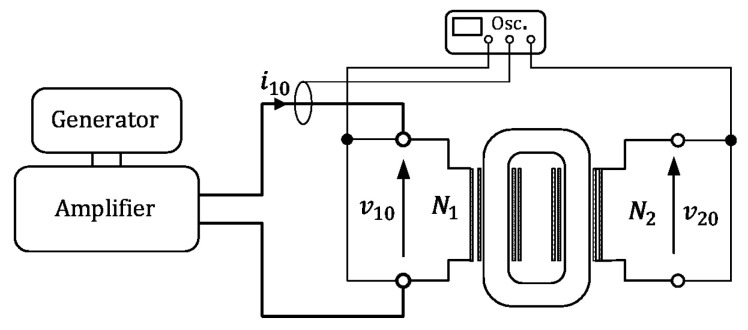
The simplified diagram for measurement of the total core losses.

**Figure 13 sensors-24-03228-f013:**
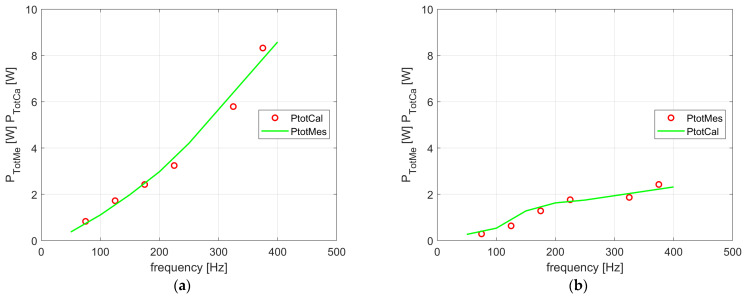
Comparison of the calculated and measured total losses: (**a**) for transformer *T*_1_; (**b**) for transformer *T*_2_.

**Table 1 sensors-24-03228-t001:** Main dimensions of the transformers (in [mm]).

Symbol	Transformer 1	Transformer 2
a	24	16
b	40	40
d	75	52
h	128.6	102
e	1.35	1.35
g	29	29

## Data Availability

Data are contained within the article. Dataset available on request from the authors.

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
