# Peer review of "Three-Dimensional Numerical Field Analysis in Transformers to Identify Losses in Tape Wound Cores"

_sensors, 2024, doi:10.3390/s24103228_

Round 1

Reviewer 1 Report

Comments and Suggestions for Authors

Dear Authors,

Thank for your manuscript.

The major issue, in my opinion, is the manuscript submission to the Sensors, since I do not fully sure its subject fully meets the scope of the Journal.

Also I have some additional comments:

1. The manuscript starts with the medium frequencies transformers of the kHz range and their actuality, as the Authors claim, "the proper values of appropriate loss must be predicted for these operating conditions when designing medium-frequency transformers...", but further the simulation and the experiments are made for 50-400 Hz range. It should be explained in more detail.

2. The text contains some facts not covered by any references, e.g.

Line 18, "These cores were mainly manufactured either as laminated or as magnetically soft composites";

Line 38, "leads to well-known analytical expression for eddy current losses determination";

Line 67, "Below are shown quite simple analytical formulas...", etc.

All the data or equations not firstly introduced by the Authors have to be properly cited.

3. The Authors claim "The soft magnetic materials are characterized by three main parameters i.e. magnetic flux density B, magnetic field intensity H and core losses P", nevertheless the magnetic permeability is also one of the most important parameter for soft magnetic materials, alongside with the critical frequency.

4. Magnetic viscosity can also be an origin of the core losses, especially for ferrites, see for example Coey JMD. Magnetism and Magnetic Materials. Cambridge University Press; 2010.

5. I recommend to attract some machine learning models, since the Authors use iterative approach for simulation. Or at least to compare the used approach with the machine learning.

6. The Authors claim the correct simulation results have been achieved after 4 passes for the first transformer and 3 passes for the second one, but I cannot find any information on the accuracy when the simulation results are compared with the experiment. The Authors only state "the very small differences between the values given in these figures validate our modeling of the losses in the laminated core", but there is no numerical data to estimate the model applicability.

Comments on the Quality of English Language

The language quality is quite poor and it needs to be checked by the native speaker before resubmitting the manuscript.

The text contains some misprints and incorrect phrases, e.g.

Line 34, "...where E – is the rms value of the electromotive force, R –is the resistance of the single 34 sheet, V – is...";

Line 40, "...where 𝜎 – is electrical conductivity of the sheet, f –is frequency of the magnetic flux 40 density, 𝐵𝑚av –is maximal average value of the flux density in the sheet, d – is thickness 41 of the sheet, V – is volume...";

Line 67, "Below are shown quite simple...";

Line 80, "Below is given a more precise analytical expression, which applied the coefficients obtained from measurements, and is related...";

Line 142, "Below are implemented, in Ansys software, the relations for...", etc.

Author Response

Dear Reviewer

Thank you for your remarks. They were taken into consideration in red color.  

Reviewer 2 Report

Comments and Suggestions for Authors

1. The introduction part needs to be significantly improved. The authors spend a lot of space trying to explain the basic concepts about soft ferromagnetic materials, which is not necessary. The authors can introduce these concepts in Section 2. However, the novelty of this paper compared to the previous research is not shown in the introduction. The literature review seems insufficient. Is there any other relevant research about numerical analysis of transformers? The authors should emphasize more on the problem they are trying to solve and the novelty of their work in the introduction instead of just introducing basic concepts.

2. The modified IHM is not well explained. “After the magnetic field analysis, we verified the value of magnetic flux density inside the core. When its estimation is correct, the value of the total loss is checked.” How do the authors verify the value of the magnetic flux? What does the “estimation” refer to?

3. The explanation of the modified IHM needs to be significantly improved. The symbols like P_TotC, B_mC, episilon_BmC, episilon_Bm, episilon_TotC, episilon_PTot, need to be explained.

4. The stopping criterion of the modified IHM process needs to be explained. What does “the absolute value of episilon_BmC smaller than episilon_Bm” mean? What does “the absolute value of episilon_PtotC smaller than episilon_PTot” mean?

5. The paper seems lacking novelty. The modified IHM seems only a data fitting process utilizing a 3D numerical simulation software, which does not exhibit enough academic innovation. 

Comments on the Quality of English Language

The English is OK. There some minor English problems such as "the modified our IHM". "In this paper are presented the results of our research concerning..."

Author Response

Dear Reviewer

Thank you for your remarks. They were taken into consideration in blue color.  

Reviewer 3 Report

Comments and Suggestions for Authors

Comments are in the file attached. 

Author Response

Thank you for your remarks. They were taken into consideration in green color.  

Round 2

Reviewer 1 Report

Comments and Suggestions for Authors

Dear Authors,

Thank you for taking into account my comments.

Reviewer 3 Report

Comments and Suggestions for Authors

The authors revised manuscript and it can be published now.